# Comparative genomic profiling of SLC26A4-expressing cells in the inner ear and other organs

**Keiji Honda**◉*, **Akimasa Kajino**◉, **Takeshi Tsutsumi**◉

Department of Otorhinolaryngology, Institute of Science Tokyo, Tokyo, Japan

* honda.oto@tmd.ac.jp (HK)

## Abstract

Pendred syndrome and autosomal recessive non-syndromic hearing loss, type 4 (DFNB4), are associated with mutations in *SLC26A4* that encodes the anion transporter SLC26A4 (pendrin). SLC26A4 is expressed in mitochondria-rich cells of the endolymphatic sac, spindle and root cells in the cochlear lateral wall, transitional cells in the vestibular organs, follicular cells in the thyroid, and type B intercalated cells in the kidney. This study aimed to assess the gene profiles of murine *Slc26a4*-expressing cells to better understand the regulatory mechanisms and functions of SLC26A4. Publicly available murine single-cell or single-nucleus RNA-sequencing (RNA-seq) datasets from the endolymphatic sac, cochlear lateral wall, utricle, kidney, airway, epididymis, and salivary glands were collected. After quality control, principal component analysis and clustering, distinct cell populations were identified, and differentially expressed genes (DEGs) were analyzed. The datasets were integrated for comparison across multiple tissues and organs. The results revealed no shared genetic profile among inner ear *Slc26a4*-expressing cells, with *Slc26a4* being the only shared DEG, suggesting that regulatory genes may include low expression transcripts, splicing variants, or long non-coding RNAs undetectable by single-cell analysis. Comparative analysis within the ionocyte family identified distinct DEGs such as *Insrr* and *Hmx2* in *Slc26a4*-expressing cells from the endolymphatic sac and kidneys, potentially significant in ion homeostasis and SLC26A4 regulation. This study highlights the specificity and complexity of SLC26A4 expression and highlights the challenges and limitations of single-cell analysis. Future research should address regulatory elements such as low-expression genes, splicing variants, and non-coding RNAs to enhance our understanding of SLC26A4 regulation across various cellular contexts.

## Introduction

*SLC26A4*, which encodes the anion transporter pendrin, is responsible for Pendred syndrome, a condition that causes congenital hearing loss, enlargement of the vestibular aqueduct (EVA), and thyroid gland enlargement (goiter), and for non-syndromic hearing loss (deafness), autosomal recessive, type 4 (DFNB4) that occurs without thyroid abnormalities [1]. Pendred syndrome/DFNB4 is the third most common genetic cause of hearing loss, with SLC26A4 mutations accounting for 6% of hereditary hearing loss cases [2]. *SLC26A4* is expressed in the

**Data availability statement:** All data files are available from the Gene Expression Omnibus (GEO) database (accession numbers GSE87293, GSE152551, GSE155966, GSE107585, GSE145443, GSE102580, and GSE150327). The R scripts used for data analysis in this study, including preprocessing, clustering, and visualization, are publicly available at Figshare (DOI: 10.6084/m9.figshare.28190282).

**Funding:** KH received funding from the Japan Society for the Promotion of Science (JSPS) KAKENHI Grant Number JP21K09602 (https://www.jsps.go.jp/english/). The funder had no role in study design, data collection and analysis, decision to publish, or preparation of the manuscript.

**Competing interests:** The authors have declared that no competing interests exist.

inner ear, specifically in mitochondria-rich cells (MRCs) of the endolymphatic sac, root cells (RoCs) and spindle cells (SpCs) of the cochlear lateral wall, and transitional cells (TCs) of the vestibule [3–5]. Studies using mouse models of human Pendred syndrome/DFNB4 have revealed that SLC26A4 dysfunction disrupts ion transport in MRCs of the endolymphatic sac, preventing normal endolymph absorption in the embryonic inner ear, which consequently leads to dilated endolymphatic space, EVA, and endolymph acidification [6,7]. These changes lead to hearing loss, potentially driven by oxidative stress, impaired cell-to-cell communication, and loss of $K^+$ secretion in the stria vascularis [8]. In humans, not all cases of EVA are associated with hearing loss, and the functional significance and disease relevance of SLC26A4 expression in RoC/SpCs and vestibular TCs of the cochlear lateral wall remain unclear.

The mechanisms regulating SLC26A4 expression are yet to be elucidated. MRCs of the endolymphatic sac and type B intercalated cells of the kidney—that are classified as ionocytes—are regulated by the forkhead box I1 (FOXI1) transcription factor for SLC26A4 expression [9–11]. However, SLC26A4 expression is absent in other ionocytes expressing FOXI1, such as kidney type A intercalated cells [10], epididymal clear cells [12], and ionocytes in the airway and salivary gland [13–15]. This indicated that additional regulatory factors may mediate the relationship between FOXI1 and SLC26A4. Similarly, RoCs, SpCs, TCs, and thyroid follicular cells express SLC26A4 independently of FOXI1, indicating the existence of alternative transcriptional regulators for SLC26A4.

This study aimed to compare the gene profiles of murine *Slc26a4*-expressing cells in the inner ear and ionocytes across multiple organs. By identifying both common and distinct functions and transcriptional regulatory mechanisms of these cells, our findings provide valuable insights into the unique roles of SLC26A4 in different tissues and their potential implications for developing novel therapeutic strategies to address Pendred syndrome, DFNB4, and other SLC26A4-related diseases.

## Methods

### Data processing

Publicly available murine single-cell or single-nucleus RNA-seq datasets were collected and used in this study (Table 1) [7,14,16–20]. Data processing and analysis were performed using the Seurat package (version 5.0.3) [21] in R (version 4.3.3) [22]. Low-quality cells were filtered out based on high mitochondrial gene content (< 10% or < 30% for the endolymphatic sac) or abnormal gene counts (< 200 or > 9,000 detected genes). For single-nucleus RNA-seq data from the lateral wall, doublet nuclei were identified and excluded using the DoubletFinder package (version 2.0.4) [23]. Subsequently, the data were normalized using Seurat's 'NormalizeData' function, using the LogNormalize method, where gene expression was scaled by total expression, multiplied by 10,000, and log-transformed.

**Table 1. Datasets used in this study.**

| Tissue | Study ID [reference] | Method | Platform | Age |
|---|---|---|---|---|
| Endolymphatic sac | GSE87293 [7] | single-cell RNA-seq | Fluidigm C1 | E12.5, E16.5, P5, P30 |
| Lateral wall | GSE152551 [16] | single-nucleus RNA-seq | 10x Genomics Chromium | P30 |
| Utricle | GSE155966 [17] | single-cell RNA-seq | FACS+Smart-seq2 | P2, P4, P6 |
| Kidney | GSE107585 [18] | single-cell RNA-seq | 10x Genomics Chromium | Adult |
| Epididymis | GSE145443 [19] | single-cell RNA-seq | 10x Genomics Chromium | 10–12 weeks |
| Airway | GSE102580 [14] | single-cell RNA-seq | inDrops | 6-8 weeks |
| Salivary gland | GSE150327 [20] | single-cell RNA-seq | 10x Genomics Chromium | E12, E14, E16, P1, P30 |

## Clustering

For each dataset, highly variable genes were identified using the'FindVariableFeatures' function using the variance stabilizing transformation method. The top 2,000 variable genes were retained for downstream analysis to capture the most dynamic expression patterns across cells. Subsequently, principal component analysis (PCA) was performed on the scaled data of the selected variable features. The significance of principal components was assessed using the 'ElbowPlot' function. The elbow method identifies the optimal number of principal components by plotting the variance explained for each component and selecting the elbow point, where adding m additional components contribute only marginally to the explained variance. Based on the elbow plot, the number of principal components used in the subsequent analyses was determined for each dataset based on the elbow plot, with values ranging from 10–50. The'FindNeighbors' function was used to compute a shared nearest neighbor graph based on the Euclidean distance in PCA space. Cell clusters were identified using the 'FindClusters'function that uses the Louvain algorithm for community detection. The resolution parameter was set between 0.5 and 1.0 for fine-grained clustering, suitable for identifying distinct cell populations. To visualize the clustered cells in two dimensions, the uniform manifold approximation and projection (UMAP) technique was applied using the'RunUMAP'function with the same principal components used in the neighbor search.

## Data integration

Data from multiple sources were integrated to minimize batch effects. After identifying highly variable genes in each dataset, the'FindIntegrationAnchors'function was used to detect anchors between datasets based on shared gene expression profiles. Subsequently, the'IntegrateData'function was applied to harmonize the datasets using these anchors. The integrated data were scaled and centered to prepare for further analysis, including dimensionality reduction and clustering.

## Differential gene expression analysis

Differentially expressed genes (DEGs) for each cell group were identified using the'FindMarkers'function with a log2 fold-change threshold of 0.5. DEGs shared by multiple cell groups were visualized using the R packages VennDiagram (version 1.7.3) [24] and ComplexHeatmap (version 2.18.0) [25]. The significantly upregulated biological processes based on the top 100 DEGs for each cell group were computed and visualized using the clusterProfiler package (v.4.10.1) [26].

## Ethics statement

This study did not involve any experimental use of animals. The analyses were performed using publicly available gene expression data, and no direct interaction with animals was conducted.

## Results

To analyze the gene profiles of *Slc26a4*-expressing cells in the inner ear, we examined publicly available single-cell datasets. The single-cell RNA-seq dataset from the endolymphatic sac [7] comprised 214 cells from wild-type (WT) mice (C57BL/6J) at embryonic days (E) 12.5, E16.5, postnatal days (P) 5, and P30. These cells were classified as MRCs, progenitor cells(ProCs), early ribosome-rich cells (RRCs), and late RRCs through clustering (Fig 1A). MRCs formed a single cluster from E16.5 to P30 and exhibited strong *Slc26a4* expression. The lateral wall

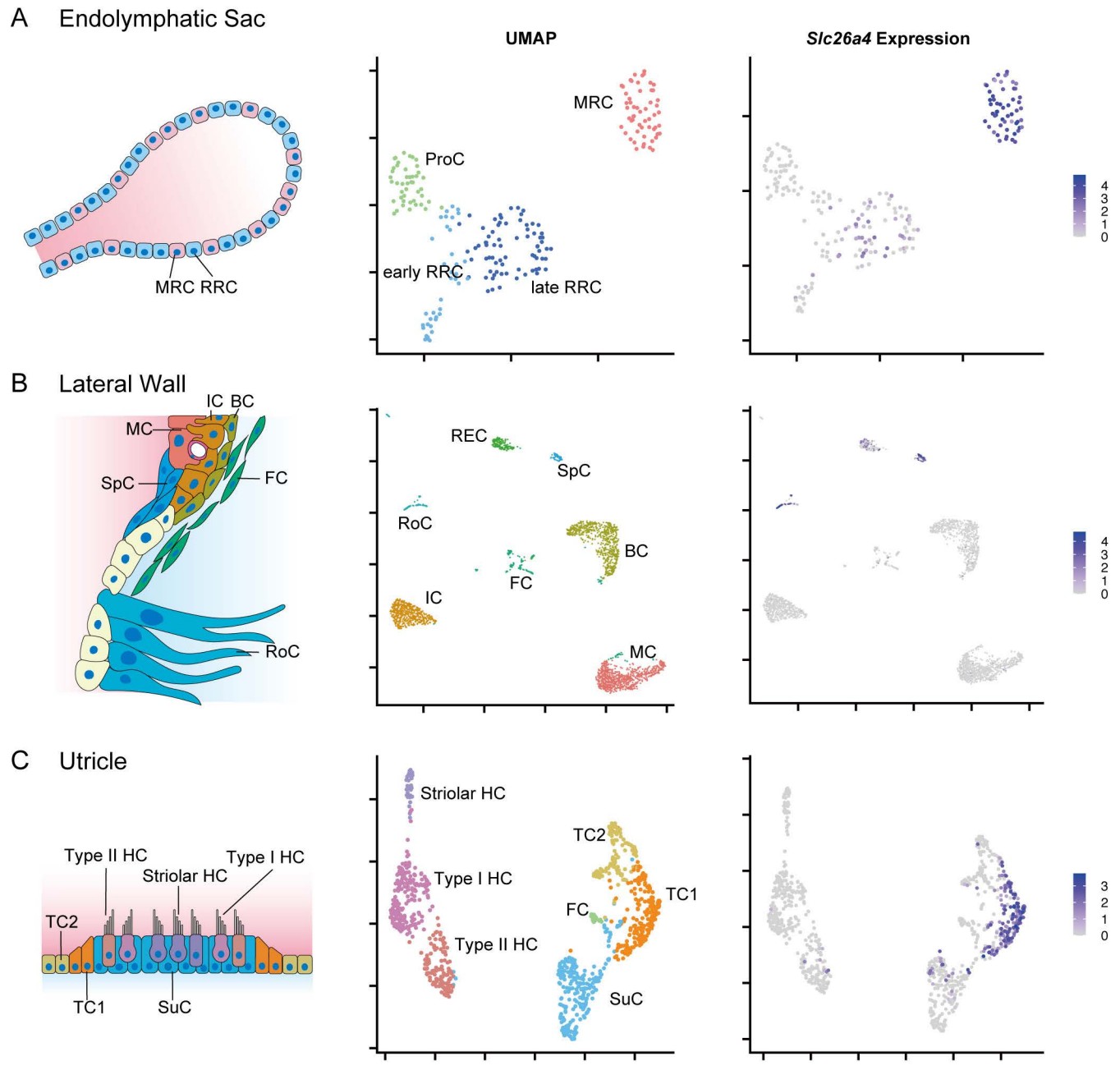

**Fig 1. Identification of *Slc26a4*-expressing cells through single-cell RNA-seq in the mouse inner ear.** This figure includes schematic illustrations, UMAP plots, and feature plots of *Slc26a4* expression across various tissues (A: endolymphatic sac, B: lateral wall, and C: utricle). Cell-type annotations adhere to the classifications observed in the original literature. The color intensity in the feature plots corresponds to the *Slc26a4* expression levels, with MRCs in the endo-lymphatic sac, SpCs and RoCs in the lateral wall, and TC1 in the utricle, forming clusters indicative of high *Slc26a4* expression. The lateral wall illustration has been adapted from a figure in the original literature.

dataset [16] was derived from single-nucleus RNA-seq data of 2502 cells collected from P30 WT mice (CBA/J). It predominantly comprised cells from the stria vascularis, such as marginal cells (MCs), intermediate cells (ICs), and basal cells (BCs), with additional contributions from Reissner membrane epithelial cells (RECs), SpCs, RoCs, and fibrocytes (FCs) (Fig 1B). *Slc26a4* expression was observed in both SpCs and RoCs. The utricle dataset [17]—comprised of 905 cells from WT mice (FVB) at P2, P4, and P6—was categorized into type I hair cells (HCs), type II HCs, striolar HCs, supporting cells (SuCs), TCs, and FCs (Fig 1C). The TCs were further divided into two types, with *Slc26a4* expression identified in TC1. Additionally, we examined datasets from the crista [27]; however, they were excluded from this analysis because they did not contain *Slc26a4*-expressing cells.

To compare the gene profiles of *Slc26a4*-expressing cells across various tissues, we integrated three distinct single-cell datasets (Fig 2). Despite variations in library preparation methods, mouse genetic backgrounds, and developmental stages across the datasets, the UMAP analysis segregated each cell group into distinct clusters (Fig 2A). The analysis of DEGs within these cell groups revealed the inclusion of canonical cell markers, confirming the cellular identity and specificity of each cluster. Additionally, the expression patterns and levels of *Slc26a4* in the integrated datasets (Figs 2B and 3C) aligned with the trends in SLC26A4 signals detected through immunostaining, in which the signals in MRCs were higher than that in SpCs, RoCs, and TCs, with weak expression also observed in RRCs [3,7]. These results highlight the robustness of the data integration process despite the inherent variations in the datasets.

Subsequently, we aimed to identify shared DEGs among *Slc26a4*-expressing cells in the inner ear, which included DEGs in MRCs (2538 genes), TC1 (1978 genes), SpCs (283 genes), and RoCs (362 genes) (Fig 3A). The only DEG shared by all four cell groups was *Slc26a4*. Transmembrane protein 176A *Tmem176a* was shared by MRCs, TC1, and SpCs; three DEGs (*2810474019Rik*, amyloid beta precursor protein *App*, and epithelial splicing regulatory protein 1 *Esrp1*) were shared by MRCs, SpCs, and RoCs; one DEG (tetraspanin 8 *Tspan8*) was shared by MRCs, TC1, and RoCs; and eight DEGs (cysteine dioxygenase type 1 *Col1*, ectonucleoside triphosphate diphosphohydrolase 3 *Entpd3*, family with sequence similarity 184 member B *Fam184b*, integrin beta-like 1 *Itgbl1*, inositol-tetrakisphosphate 1-kinase *Itpk1*, platelet-derived growth factor C *Pdgfc*, tropomodulin 1 *Tmod1*, and transmembrane protein 132C *Tmem132c*) were shared by TC1, SpCs, and RoCs. Assessing the expression levels of these genes across each cell group revealed no patterns similar to those of *Slc26a4* (Fig 3B). To assess the processes and functions of the shared genes among *Slc26a4*-expressing cells, we conducted a gene ontology enrichment analysis using the top 100 DEGs from MRCs, TC1, SpCs, and RoCs. However, no biological processes were shared between the three or more cell groups (Fig 3C). These results indicated that within the scope of single-cell dataset-detected genes, there was no shared gene profile among *Slc26a4*-expressing cells in the inner ear.

Therefore, we aimed to explore genes shared among *Slc26a4*-expressing cells, including those from various organs beyond the inner ear. *Slc26a4*-expressing cells outside the inner ear include thyroid follicular cells and type B intercalated cells of the kidney. We failed to identify *Slc26a4*-expressing cell groups in two available mouse thyroid datasets, likely because *Slc26a4* expression levels were below the detection threshold of single-cell RNA-seq. Therefore, we focused on genes expressed in MRCs and type B intercalated cells, but not in other ionocytes. Single-cell datasets of the kidneys, airway (trachea), epididymis, and salivary glands were collected and compared. Clustering within each dataset identified specific clusters expressing the ionocyte marker *Foxi1*, annotated as airway ionocytes, epididymis clear cells, salivary gland ionocytes, and intercalated cells (Fig 4A). Intercalated cells were further categorized into two clusters, type A and B, based on the expression patterns of canonical markers, such as *Slc26a4*,

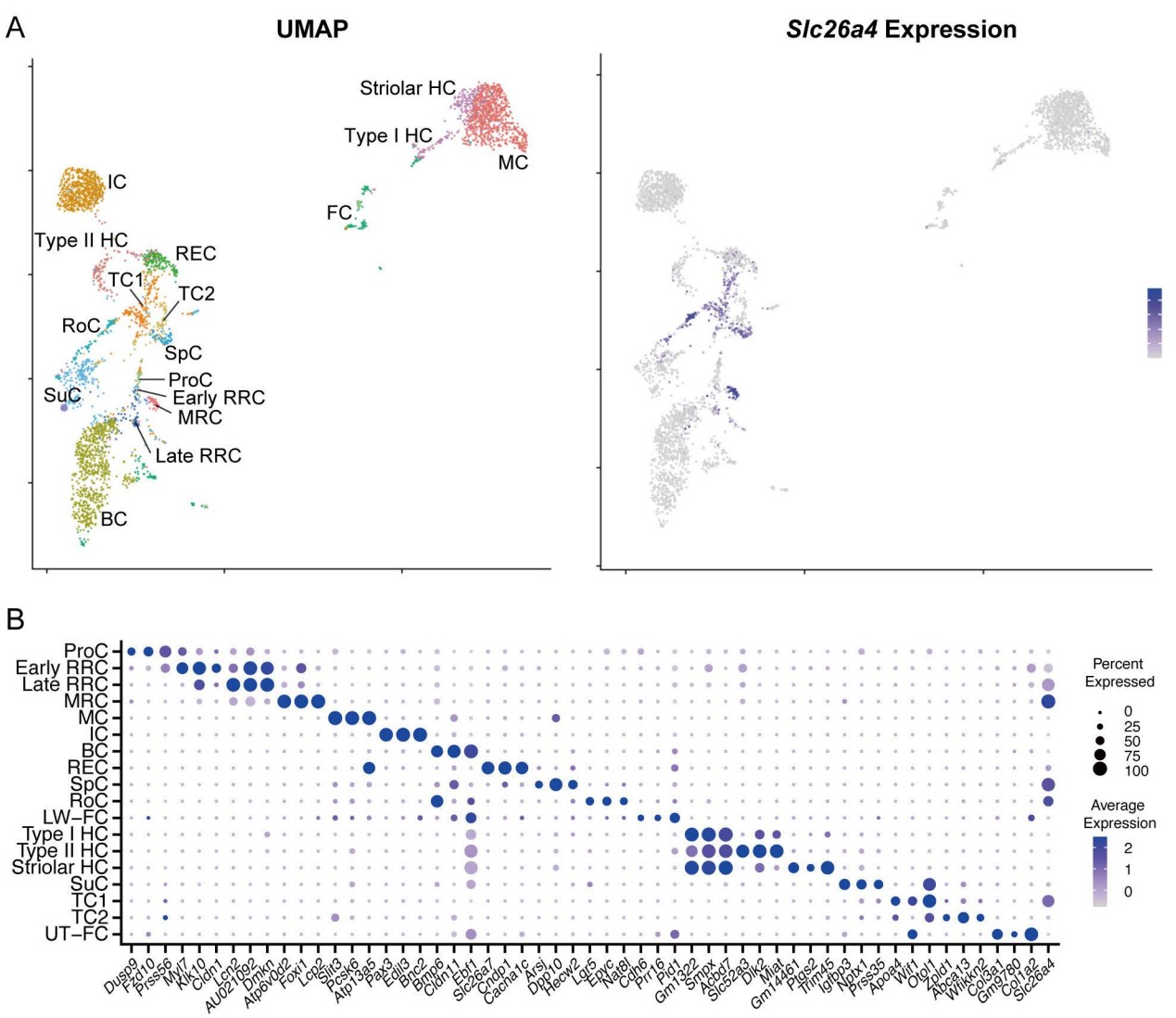

**Fig 2. Integrated expression analysis of *Slc26a4* in the mouse inner ear, combining datasets from the tissues examined in** Fig 1**.** A: The UMAP plot presents a comprehensive view of cell type distribution and *Slc26a4* expression levels within the mouse inner ear. The feature plot on the right details *Slc26a4* expression across the cell types. B: Dot plot of the top three DEGs for each cell type and *Slc26a4*; dot size and color intensity represent the percentage of cells expressing a given gene and the average expression levels among expressing cells, respectively. The x-axis lists the genes, whereas the y-axis categorizes the cell types from the integrated tissue datasets.

*Slc4a9*, and *Slc4a1*. *Slc26a4* was specifically expressed in MRCs and type B intercalated cells (Fig 4B). Subsequently, the DEGs of ionocytes from each organ were identified and explored for genes common to each group. There were 68 DEGs commonly expressed across all ionocytes, including previously reported ionocyte markers, such as *Foxi1*, ATPase H$^+$ transporting V0 subunit D2 *Atp6v0d2*, and transcription Factor CP2 like 1 *Tfcp2l1* (Fig 4C). Additionally, 46 DEGs were unique to MRCs and type B intercalated cells (Fig 4C). Of these, 14 genes exhibited higher average expression levels in MRCs and type B intercalated cells than that in other cells, with expression patterns similar to those of *Slc26a4* (Fig 4D). These genes are candidates for involvement in the ionocyte functions associated with *SLC26A4* and their regulatory mechanisms.

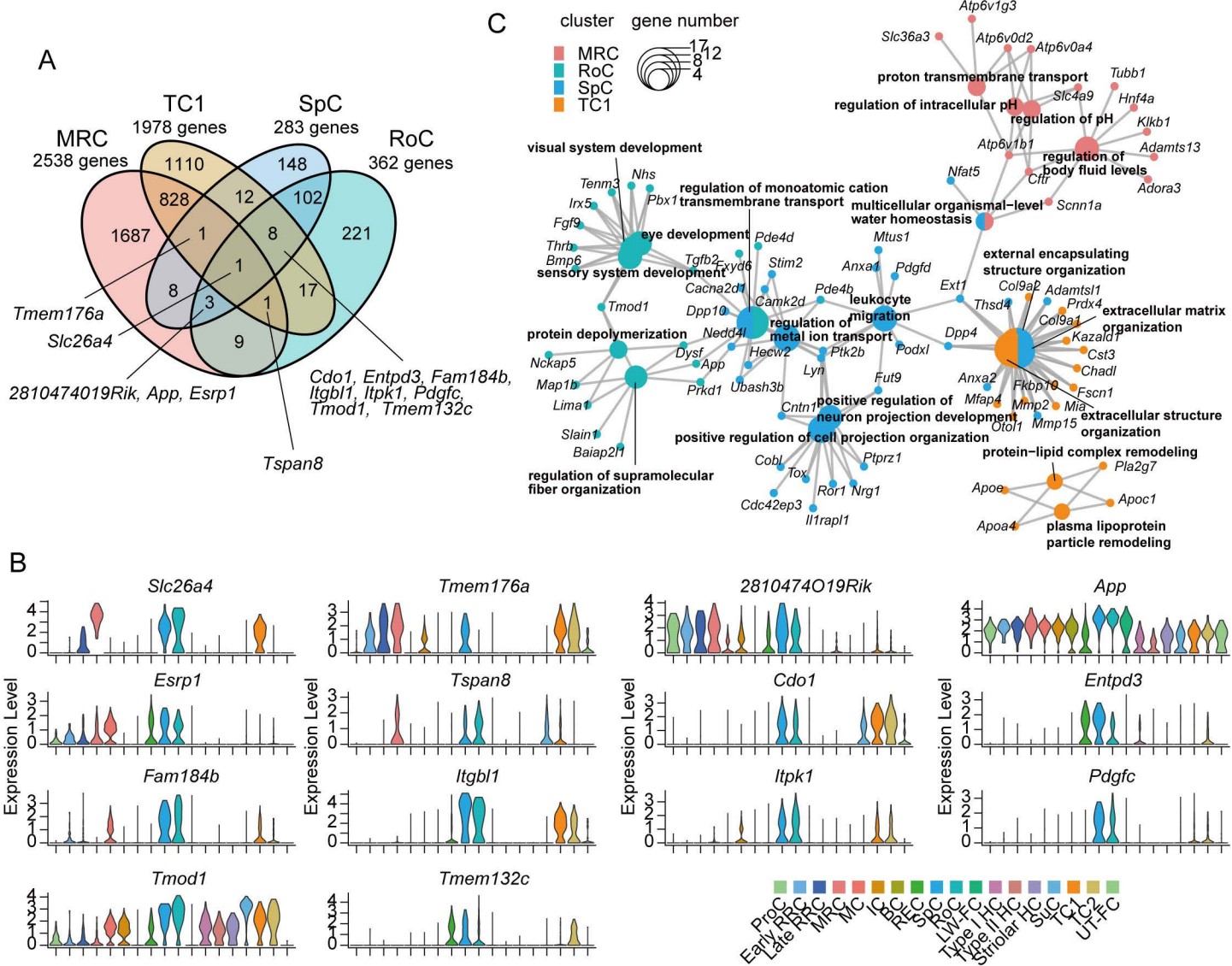

**Fig 3. Identification of distinctive marker genes and functional similarities among *Slc26a4*-expressing cells in the mouse inner ear.** A: Venn diagram illustrates the number of DEGs unique to and shared between the four cell types. B: Violin plots provide the distribution patterns of expression levels for the DEGs shared by all or subsets of the *SLC26A4*-expressing cells. C: The network graph displays the enriched biological processes of the Gene Ontology terms associated with the top 100 DEGs within these cells, with node size indicating gene count and color indicating specific cell types.

## Discussion

In this study, we used publicly available single-cell or single-nucleus RNA-seq datasets to identify marker genes that are commonly expressed in *Slc26a4*-expressing cells across various tissues. Despite thorough analysis, no markers were universally expressed in the MRCs of the endolymphatic sac, RoCs and SpCs of the lateral wall, and TCs of the utricle. However, comparative analysis within the ionocyte family revealed numerous markers that are commonly expressed in the MRCs and type B intercalated cells of the kidney, which also express *Slc26a4*.

One of the marker genes for *Slc26a4*-expressing ionocytes, insulin receptor-related receptor (*Insrr*), encodes insulin receptor-related receptor (IRR) [28]. IRR is a member of the insulin

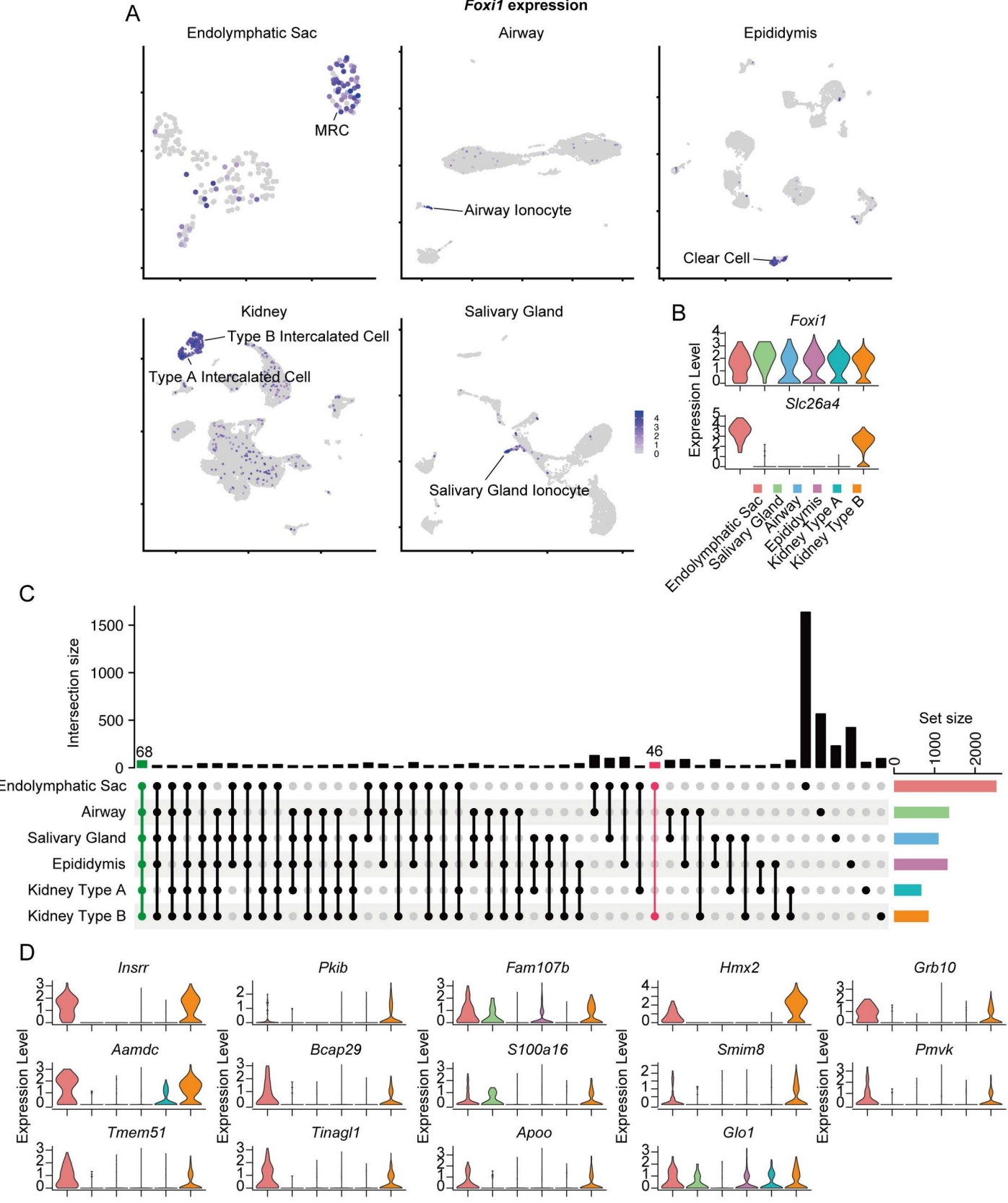

**Fig 4. Shared marker gene identification in *Slc26a4*-expressing ionocytes across various organs.** A: UMAP plots illustrate the specific expression of *Foxi1*, a hallmark transcription factor for ionocytes, in various organs: endolymphatic sac (MRCs), airway (ionocytes), epididymis (clear cells), salivary glands (ionocytes),

and kidneys (type A and B intercalated cells). B: Violin plots demonstrate the comparative expression levels of *Foxi1* and *Slc26a4* within these ionocytes. *Slc26a4* is distinctively expressed in MRCs of the endolymphatic sac and type B intercalated cells of the kidney. C: An upset plot visualizes the overlap of DEGs among the ionocyte populations. Vertical bars indicate the number of shared DEGs among various cell group combinations. Horizontal bars indicate the total DEGs in each group. A total of 68 genes (highlighted in green), including *Foxi1,* are shared across all ionocyte groups, which are ionocyte markers. Additionally, a distinct subset of 46 genes (highlighted in pink), including *Slc26a4*, are shared between MRCs and type B intercalated cells. D: Violin plots reveal the expression patterns of DEGs shared between MRCs and type B intercalated cells, indicating their potential roles in the regulation and functionality of *Slc26a4*.

receptor family that includes the insulin receptor (IR) and insulin-like growth factor receptor. Unlike other family members, IRR is uniquely activated under alkaline conditions and is believed to serve as an extracellular alkali sensor [29]. IRR is specifically expressed in type B intercalated cells of the kidneys [30]. RNA-seq analysis of the kidneys from *Insrr* knockout mice revealed reduced *SLC26A4* expression compared to that in WT controls [31]. In the inner ear, IRR may also detect an alkaline environment surrounding MRCs and facilitate *Slc26a4* expression to secrete bicarbonate ions into the endolymphatic space.

Another gene identified as a marker of *Slc26a4*-expressing ionocytes is the H6 family homeobox 2 gene (*Hmx2*). *Hmx2* is essential for the normal inner ear development. Disruptions in *Hmx2* result in abnormal vestibular development, including reduced cell proliferation, absence of semicircular ducts, and loss of vestibular sensory patches [32,33]. Hemizygous deletions of *HMX2* and *HMX3* in humans are associated with inner ear malformations, vestibular dysfunction, and congenital sensorineural hearing loss [34]. Single-cell data from the endolymphatic sac revealed that *Hmx2* was expressed in MRCs, progenitor cells, and RRCs during embryonic stages; however, its expression was restricted to MRCs as development progressed. Although *Hmx2* has no reported role in the kidneys, the single-cell data revealed its exclusive expression in type B intercalated cells, with no expression in type A intercalated cells or other kidney cells. This finding indicated a potential association between *Hmx2* and *Slc26a4* expression.

This study did not identify any marker genes that are commonly expressed in *Slc26a4*-expressing cells of the inner ear. Numerous factors may contribute to this outcome. First, RNA detection sensitivity is a challenge in single-cell analysis because low-expression genes, including those co-expressed with *Slc26a4*, may be unidentified. Additionally, droplet-based single-cell analysis methods, such as chromium or inDrops, solely analyze the 3' ends of RNA, which may overlook splice variants. Moreover, unknown cell-specific long non-coding RNAs (lncRNAs) that are not mapped to the reference genome may be involved in the transcription and function of *Slc26a4*. For example, in MRCs of the endolymphatic sac, numerous lncRNAs located adjacent to *Foxi1* on the chromosome are specifically expressed [7].

Two proteins have been previously reported to interact with SLC26A4. One of these, the IQ-motif containing GTPase-activating protein 1 (IQGAP1), was discovered through a yeast two-hybrid screen using a mouse kidney cDNA library with the human SLC26A4 C-terminal region serving as bait [35]. IQGAP1 is localized alongside SLC26A4 on the apical membrane of type B intercalated cells, suggesting its involvement in modulating SLC26A4 trafficking and functionality. Another interacting protein, the μ2 subunit of the adaptor protein 2 (AP-2) complex, was identified through both yeast two-hybrid and pull-down assays [36]. As a key component in clathrin-mediated endocytosis, the μ2 subunit controls the abundance of SLC26A4 at the apical membrane of MRCs in the endolymphatic sac. The datasets from the present study demonstrated that the genes encoding the two proteins, *Iqgap1* and *Ap2m1*, are not specifically expressed in *Slc26a4*-expressing cells but instead exhibit ubiquitous expression across different cell types. This widespread expression at the gene level indicates that their roles in modulating SLC26A4 are likely context-dependent, potentially relying on factors such as cell-type-specific signaling pathways, the presence of cofactors, or the unique structural features of SLC26A4-expressing cells. This highlights the complexity of SLC26A4 regulation and

the importance of further investigation into the molecular and cellular contexts that enable these interactions to selectively influence its function.

A limitation of this study is the variability in library preparation methods, mouse backgrounds, and age, which may have introduced batch effects and affected the results despite efforts to mitigate these through anchor-based integration. While the integration process effectively reduces batch effects, residual variability may persist, potentially affecting the interpretation of certain findings. Future studies can ideally confirm these results by performing single-cell RNA-seq on samples collected from genetically identical mice of the same age and under standardized conditions.

Future research could build upon this study by focusing on two main directions. Pooling Slc26a4-expressing cells for bulk RNA-seq analysis may provide a more comprehensive understanding of their transcriptomic landscape, enabling the detection of low-expressed RNAs, splicing variants, and long non-coding RNAs (lncRNAs). Alternatively, the use of more sensitive single-cell RNA-seq methods, such as full-length transcript sequencing technologies, could improve the detection of low-abundance transcripts and enhance our understanding of the heterogeneity and specific functions of *Slc26a4*-expressing cells.

## Conclusion

This study assessed the transcriptional profiles of *Slc26a4*-expressing cells across various tissues and identified the primary genes that may play significant roles in specific ionocytes. However, no universal markers for *Slc26a4*-expressing cells in the inner ear were identified, highlighting the challenges and limitations of single-cell analysis. Future studies should focus on enhancing detection technologies and exploring the roles of low-expression genes, splicing variants, and non-coding RNAs to better understand *Slc26a4* regulation in various cellular environments.

## Acknowledgement

We sincerely thank Yoshiyuki Kawashima and Taku Ito for their invaluable feedback and constructive review of this manuscript. We would like to thank Editage (www.editage.jp) for English language editing.

## Author contributions

**Data curation:** Keiji Honda.

**Formal analysis:** Keiji Honda.

**Funding acquisition:** Keiji Honda.

**Investigation:** Keiji Honda, Akimasa Kajino.

**Methodology:** Keiji Honda.

**Project administration:** Keiji Honda.

**Resources:** Keiji Honda.

**Software:** Keiji Honda.

**Supervision:** Takeshi Tsutsumi.

**Validation:** Keiji Honda, Akimasa Kajino.

**Visualization:** Keiji Honda, Akimasa Kajino.

**Writing – original draft:** Keiji Honda.

**Writing – review & editing:** Keiji Honda, Akimasa Kajino, Takeshi Tsutsumi.

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
