## [Decision Letter · Decision Letter 0]

15 Dec 2024

PONE-D-24-55010Comparative genomic profiling of SLC26A4-expressing cells in the inner ear and other organsPLOS ONE

Dear Dr. Honda,

Thank you for submitting your manuscript to PLOS ONE. After careful consideration, we feel that it has merit but does not fully meet PLOS ONE’s publication criteria as it currently stands. Therefore, we invite you to submit a revised version of the manuscript that addresses the points raised during the review process.

We look forward to receiving your revised manuscript.

Kind regards,

Nejat Mahdieh

Academic Editor

PLOS ONE

Journal Requirements:

3. Thank you for stating the following in the Acknowledgments Section of your manuscript: This work was supported by JSPS KAKENHI Grant Number JP21K09602. We sincerely than k Yoshiyuki Kawashima and Taku Ito for their invaluable feedback and constructive review of this manuscript. We would like to thank Editage (www.editage.jp) for English language editing

Please remove any funding-related text from the manuscript and let us know how you would like to update your Funding Statement. Currently, your Funding Statement reads as follows: KH received funding from the Japan Society for the Promotion of Science (JSPS) KAKENHI Grant Number JP21K09602 (https://www.jsps.go.jp/english/). The funder had no role in study design, data collection and analysis, decision to publish, or preparation of the manuscript.

Reviewers' comments:

Reviewer's Responses to Questions

**Comments to the Author**

1. Is the manuscript technically sound, and do the data support the conclusions?

Reviewer #1: Yes

Reviewer #2: Yes

2. Has the statistical analysis been performed appropriately and rigorously? 

Reviewer #1: Yes

Reviewer #2: Yes

3. Have the authors made all data underlying the findings in their manuscript fully available?

Reviewer #1: Yes

Reviewer #2: Yes

4. Is the manuscript presented in an intelligible fashion and written in standard English?

Reviewer #1: Yes

Reviewer #2: Yes

5. Review Comments to the Author

Reviewer #1: This article examines the association between Pendred syndrome and autosomal recessive nonsyndromic hearing loss type 4 (DFNB4) with SLC26A4 gene mutations. The aim of this study was to evaluate the gene profiles of Slc26a4-expressing cells in mice to better understand the regulatory mechanisms and functions of this gene.

The following suggestions are made for improvement:

1- Abstract

- It is recommended to highlight the key findings in the abstract and to state them more precisely.

2- Introduction

- Statement of the objectives and applications of the research:

At the end of the introduction, it is recommended to explain what new information this research may contribute to our current knowledge.

3- Discussion

- The discussion is the most important part of an article, so please consider the following to improve it:

- Although some relevant sources are included in the discussion, referencing existing past research can enhance the quality of the discussion.

- It is recommended to compare the results of your study with those of other studies and to present hypotheses regarding the consistency or inconsistency of the evaluation results.

- Finally, while the limitations of this study are mentioned, which is very good, provide suggestions for future studies to other researchers.

Reviewer #2: The introduction addresses an important genetic factor related to hearing loss and associated syndromes, which is relevant to both clinical and research communities. The introduction sets the stage for understanding the significance of investigating SLC26A4 in different cell types and organs. However, there are some areas to improvement: Additional details on the mechanisms by which SLC26A4 dysfunction leads to hearing loss could enhance the reader's understanding of its importance. In addition, including prevalence statistics for Pendred syndrome and DFNB4 would strengthen the case for why this research is significant and necessary.

In conclusion, while it is a suitable introduction, refining it based on the points mentioned would significantly enhance its quality and impact.

Regarding methods, it is undeniable fact that all parts are acceptable and ethics statement is clear and concise, appropriately indicating that no live animals were involved in the study. This transparency is commendable. however, there are some constrictive notes: The explanation of the elbow method for determining the number of principal components is appropriate, but including a brief discussion on how this method works could enhance clarity for readers less familiar with PCA. The use of the Louvain algorithm for community detection is a solid choice, and specifying the range for the resolution parameter adds valuable detail. The integration of datasets to minimize batch effects is well-described, highlighting the use of anchor detection and harmonization methods. However, it may be beneficial to discuss any potential limitations or assumptions made during this integration process, as batch effects can sometimes persist despite these efforts.

The methods and results sections are generally well-structured and informative. They provide a comprehensive overview of the analytical approaches used in this study while maintaining clarity. Suggestions for improvement include providing additional context or rationale for certain methodological choices, discussing potential limitations, and elaborating on key findings' implications. Overall, this article contributes valuable insights into the gene expression landscape of Slc26a4-expressing cells in the inner ear

6. PLOS authors have the option to publish the peer review history of their article (what does this mean? ). If published, this will include your full peer review and any attached files.

**Do you want your identity to be public for this peer review?** For information about this choice, including consent withdrawal, please see our Privacy Policy .

Reviewer #1: No

Reviewer #2: No

---

## [Author Response · Author response to Decision Letter 0]

16 Jan 2025

Reviewer #1

This article examines the association between Pendred syndrome and autosomal recessive nonsyndromic hearing loss type 4 (DFNB4) with SLC26A4 gene mutations. The aim of this study was to evaluate the gene profiles of Slc26a4-expressing cells in mice to better understand the regulatory mechanisms and functions of this gene.

The following suggestions are made for improvement:

1. Abstract: It is recommended to highlight the key findings in the abstract and to state them more precisely.

Thank you for your suggestion. We have revised the abstract to emphasize the key findings more explicitly, focusing on the novel insights provided by our study, while ensuring it remains within the 300-word limit.

2. Introduction: At the end of the introduction, it is recommended to explain what new information this research may contribute to our current knowledge.

We have revised the last paragraph at the end of the Introduction section outlining the novel contributions of our study, particularly how it advances the understanding of the role of SLC26A4 in various tissues and the development of novel therapeutic strategies for Pendred syndrome/DFNB4 or other SLC26A4 related diseases.

3. Discussion: The discussion is the most important part of an article, so please consider the following to improve it:

- Although some relevant sources are included in the discussion, referencing existing past research can enhance the quality of the discussion. It is recommended to compare the results of your study with those of other studies and to present hypotheses regarding the consistency or inconsistency of the evaluation results.

Thank you for your constructive feedback. In response, we have added a new paragraph to the Discussion section that references previous studies on proteins interacting with SLC26A4, specifically IQGAP1 and the μ2 subunit of the AP-2 complex. These proteins have been reported to influence SLC26A4 trafficking and functionality through distinct mechanisms, as demonstrated in earlier yeast two-hybrid and pull-down assays. By incorporating data from the current study, we highlight the ubiquitous expression patterns of Iqgap1 and Ap2m1 and propose that their roles in modulating SLC26A4 are likely context-dependent, influenced by cell-specific signaling pathways and cofactors. This addition provides a comparative perspective that underscores the complexity of SLC26A4 regulation and aligns with your recommendation to enhance the discussion with relevant past research and hypotheses.　We believe this addition strengthens the Discussion section and addresses your comment effectively.

- Finally, while the limitations of this study are mentioned, which is very good, provide suggestions for future studies to other researchers.

we have added a new paragraph to the Discussion section that outlines two potential directions for future research. These include pooling Slc26a4-expressing cells for bulk RNA-seq analysis to better understand their transcriptomic landscape and utilizing more sensitive single-cell RNA-seq methods to detect low-abundance transcripts and explore cellular heterogeneity. We hope this addition provides valuable guidance for future studies.

Reviewer #2

The introduction addresses an important genetic factor related to hearing loss and associated syndromes, which is relevant to both clinical and research communities. The introduction sets the stage for understanding the significance of investigating SLC26A4 in different cell types and organs. However, there are some areas to improvement: Additional details on the mechanisms by which SLC26A4 dysfunction leads to hearing loss could enhance the reader's understanding of its importance. In addition, including prevalence statistics for Pendred syndrome and DFNB4 would strengthen the case for why this research is significant and necessary.　In conclusion, while it is a suitable introduction, refining it based on the points mentioned would significantly enhance its quality and impact.

We sincerely thank the reviewer for their thoughtful and constructive feedback on the Introduction section. We have expanded the Introduction to include a detailed explanation of the biological mechanisms linking SLC26A4 dysfunction to hearing loss, referencing a 2011 review that a foundational hypothesis for these mechanisms. We also have incorporated prevalence data for Pendred syndrome and DFNB4 to emphasize the clinical significance and relevance of our study.

Regarding methods, it is undeniable fact that all parts are acceptable and ethics statement is clear and concise, appropriately indicating that no live animals were involved in the study. This transparency is commendable. however, there are some constrictive notes:

- The explanation of the elbow method for determining the number of principal components is appropriate, but including a brief discussion on how this method works could enhance clarity for readers less familiar with PCA. The use of the Louvain algorithm for community detection is a solid choice, and specifying the range for the resolution parameter adds valuable detail.

We have added a brief explanation of how the elbow method works in the Methods section to improve clarity for readers less familiar with PCA.

- The integration of datasets to minimize batch effects is well-described, highlighting the use of anchor detection and harmonization methods. However, it may be beneficial to discuss any potential limitations or assumptions made during this integration process, as batch effects can sometimes persist despite these efforts.

In the Discussion section, we have expanded the limitations paragraph to address potential limitations and assumptions related to the integration process and batch effects. Specifically, we discuss how residual variability may persist despite anchor-based harmonization and highlight the need for standardized conditions in future studies to further minimize these effects.

The methods and results sections are generally well-structured and informative. They provide a comprehensive overview of the analytical approaches used in this study while maintaining clarity. Suggestions for improvement include providing additional context or rationale for certain methodological choices, discussing potential limitations, and elaborating on key findings' implications. Overall, this article contributes valuable insights into the gene expression landscape of Slc26a4-expressing cells in the inner ear.

Thank you for your positive and encouraging feedback on the methods and results sections. We appreciate your thoughtful suggestions, which have guided our revisions and improved the overall quality of the manuscript.

Editorial Office Comments

Thank you for the reminder regarding PLOS ONE's style requirements. We have updated the manuscript file name and confirmed that the manuscript adheres to the journal’s style guidelines. Additionally, all figure file names have been updated, and the figures have been converted to TIFF format. We also adjusted the size and font size within the figures to ensure they meet the journal’s specifications.

Thank you for highlighting the importance of code sharing to support reproducibility and reuse. In response, we have uploaded the R scripts used for data analysis in this study, including preprocessing, clustering, and visualization, to Figshare. The code is publicly available at Figshare with the DOI: 10.6084/m9.figshare.28190282. Additionally, we have added the following statement to the data availability statement:

“The R scripts used for data analysis in this study, including preprocessing, clustering, and visualization, are publicly available at Figshare (DOI: 10.6084/m9.figshare.28190282).”

3. Thank you for stating the following in the Acknowledgments Section of your manuscript: This work was supported by JSPS KAKENHI Grant Number JP21K09602. We sincerely thank Yoshiyuki Kawashima and Taku Ito for their invaluable feedback and constructive review of this manuscript. We would like to thank Editage (www.editage.jp) for English language editing.

Please remove any funding-related text from the manuscript and let us know how you would like to update your Funding Statement. Currently, your Funding Statement reads as follows: KH received funding from the Japan Society for the Promotion of Science (JSPS) KAKENHI Grant Number JP21K09602 (https://www.jsps.go.jp/english/). The funder had no role in study design, data collection and analysis, decision to publish, or preparation of the manuscript.

We have removed the funding information from the Acknowledgments section and ensured it is only included in the Funding Statement section. The funding statement does not need to be changed.

We have removed the phrase “data not shown” in the manuscript.

---

## [Decision Letter · Decision Letter 1]

26 Jan 2025

Comparative genomic profiling of SLC26A4-expressing cells in the inner ear and other organs

PONE-D-24-55010R1

Dear Dr. Honda,

We’re pleased to inform you that your manuscript has been judged scientifically suitable for publication and will be formally accepted for publication once it meets all outstanding technical requirements.

Kind regards,

Nejat Mahdieh

Academic Editor

PLOS ONE

Additional Editor Comments (optional):

Reviewers' comments:

Reviewer's Responses to Questions

**Comments to the Author**

1. If the authors have adequately addressed your comments raised in a previous round of review and you feel that this manuscript is now acceptable for publication, you may indicate that here to bypass the “Comments to the Author” section, enter your conflict of interest statement in the “Confidential to Editor” section, and submit your "Accept" recommendation.

Reviewer #1: All comments have been addressed

2. Is the manuscript technically sound, and do the data support the conclusions?

Reviewer #1: Yes

3. Has the statistical analysis been performed appropriately and rigorously? 

Reviewer #1: Yes

4. Have the authors made all data underlying the findings in their manuscript fully available?

Reviewer #1: Yes

5. Is the manuscript presented in an intelligible fashion and written in standard English?

Reviewer #1: Yes

6. Review Comments to the Author

Reviewer #1: (No Response)

7. PLOS authors have the option to publish the peer review history of their article (what does this mean? ). If published, this will include your full peer review and any attached files.

**Do you want your identity to be public for this peer review?** For information about this choice, including consent withdrawal, please see our Privacy Policy .

Reviewer #1: No

---

## [Editor Report · Acceptance letter]

PONE-D-24-55010R1

PLOS ONE

Dear Dr. Honda,

I'm pleased to inform you that your manuscript has been deemed suitable for publication in PLOS ONE. Congratulations! Your manuscript is now being handed over to our production team.

Kind regards,

on behalf of

Dr. Nejat Mahdieh

Academic Editor

PLOS ONE